# Development of an Improved Sulfur-Oxidizing Bacteria-Based Ecotoxicity Test for Simple and Rapid On-Site Application

**DOI:** 10.3390/toxics11040352

**Published:** 2023-04-08

**Authors:** Heonseop Eom

**Affiliations:** Department of Civil Engineering, Keimyung University, 1095 Dalgubeol-daero, Dalseo-gu, Daegu 42601, Republic of Korea; heom@kmu.ac.kr; Tel.: +82-53-580-5706

**Keywords:** ecotoxicity, sulfur-oxidizing bacteria, electrical conductivity

## Abstract

Microbial toxicity tests are considered efficient screening tools for the assessment of water contamination. The objective of this study was to develop a sulfur-oxidizing bacteria (SOB)-based ecotoxicity test with high sensitivity and reproducibility for simple and rapid on-site application. To attain this goal, we developed a 25 mL vial-based toxicity kit and improved our earlier SOB toxicity test technique. The current study applied a suspended form of SOB and shortened the processing time to 30 min. Moreover, we optimized the test conditions of the SOB toxicity kit in terms of initial cell density, incubating temperature, and mixing intensity during incubation. We determined that 2 × 10^5^ cells/mL initial cell density, 32 °C incubating temperature, and 120 rpm mixing intensity are the optimal test conditions. Using these test conditions, we performed SOB toxicity tests for heavy metals and petrochemicals, and obtained better detection sensitivity and test reproducibility, compared to earlier SOB tests. Our SOB toxicity kit tests have numerous advantages, including a straightforward test protocol, no requirement of sophisticated laboratory equipment, and no distortion of test results from false readings of end-points and properties of test samples, making it suitable for simple and rapid on-site application.

## 1. Introduction

There are approximately 150,000 different chemicals in commercial use and their number and applications continue to grow [1,2]. Chemicals used in homes and diverse industries are released directly or indirectly into water systems [1,2]. Although chemicals can show unknown adverse effects on the environment and ecosystems, the presence of chemicals in the water environment does not necessarily represent a risk [1]. Chemicals exceeding levels of concern cause water contamination and pose threats to aquatic ecosystems and public health [1,3].

Conventionally, water contamination has been assessed based on physicochemical quantitative analyses of water quality parameters, including dissolved oxygen, solids, biochemical or chemical oxygen demand, various nutrients, and selected contaminants [4,5,6,7]. These physicochemical analyses are useful to understanding the fundamental properties of water and obtaining detailed quantitative information of specific contaminants [8,9,10]. However, such water quality evaluation is unable to reveal the biochemical effects of contamination on living organisms and the environment [9,11]. Moreover, physicochemical quantification usually requires advanced analytic equipment, skilled personnel, lengthy processing time, and high experimental expense, making it unsuitable for on-site simple and rapid toxicity screening of contaminated water [2,12].

As supplements or alternative to physicochemical quantitative analyses, biological tests that employ organisms such as invertebrates, fish, daphnia, and microorganisms, have been widely used in toxicity assessment of contaminated water [4,13,14,15,16,17,18]. Because biological tests, generally named ecotoxicity tests, evaluate toxicity based on changes in the response of organisms to contaminants, they can directly demonstrate the impacts of contaminants on living organisms and the environment [8,15,19,20,21]. Among the diverse trophic levels of organisms, microorganism-based tests are considered particularly efficient tools for routine toxicity evaluation because they provide easy test protocols, relatively short test time, cost effectiveness, and less ethical responsibility compared to other organism-based tests [21,22,23]. Moreover, as microorganisms have diverse ecological functions, microorganism-based tests can provide important toxicological information on ecosystems. Numerous microorganisms, including bioluminescent bacteria, nitrifying bacteria, oligotrophic bacteria, *Escherichia coli*, and microalgae, have been employed in ecotoxicity tests with measurement of luminescence, growth and respiration rate, turbidity, and photosynthesis, showing favorable performance in assessing toxicity of various inorganic and organic contaminants in water [24,25,26,27,28].

In our earlier studies, we demonstrated the application of sulfur-oxidizing bacteria (SOB), specifically *Acidithiobacillus caldus*, to toxicity tests of contaminated water [2,21,29,30,31,32,33,34,35]. SOB are chemolithoautotrophic and acidophilic bacteria ubiquitous in diverse environments, including the hydrosphere [35,36]. SOB gain energy from aerobic oxidation of sulfur and produce sulfate and hydrogen ions as by-products (Equation (1)) [21,37].
S^0^ + H_2_O + 1.5O_2_ → SO_4_^2−^ + 2H^+^, ∆G^◦′^ = −587 kJ/reaction(1)

Because electrical conductivity represents the ability to carry a current and is proportional to the concentration of ions, electrical conductivity is able to serve as a proxy for the microbial activity of SOB [21,35,38]. In the presence of contaminants, SOB activity is inhibited, resulting in less generation of sulfate and hydrogen ions. Hence, SOB tests evaluate the toxicity of contaminated water by comparing increases in electrical conductivity between test samples and the control (where no contaminants exist). Our earlier studies confirmed that this SOB test is a reliable toxicity-screening technique [21,29,30,31,32,33,34,35]. SOB tests showed favorable results from toxicity assessment of heavy metals, endocrine-disrupting compounds, inorganic nitrogen, and petrochemicals in water [2,21,29,30,31,32,33,34,35]. Moreover, SOB tests have the advantages of simple test methodology and low cost, and they do not require sophisticated instruments to measure microbial activity.

The objective of this study was to develop improved SOB toxicity tests for simple and rapid on-site application. To attain this goal, we used a suspended form of SOB in this study. We expected that this approach would be more advantageous to the application of identical amounts of SOB and making direct contact with contaminants. Moreover, for better field application, the present study developed a kit-type SOB test and decreased the processing time from several hours to 30 min. In addition, we optimized test conditions such as initial cell density, incubating temperature, and mixing intensity, yielding enhanced detection sensitivity and test reproducibility. As a result, our current SOB test represents an improvement in simple and rapid on-site toxicity assessment.

## 2. Materials and Methods

### 2.1. SOB Strain and Cultivation

In the current study, a specific SOB strain, *Acidithiobacillus caldus*, was employed as the test organism. The SOB were obtained from Kangwon National University (Chuncheon, Republic of Korea) and cultivated in a liquid medium in a 500 mL conical glass flask. The medium for SOB was prepared according to Johnson et al. (1987) [39] and Duquesne et al. (2003) [40]. The medium had 0.5 g MgSO_4_∙7H_2_O, 3 g (NH_4_)_2_SO_4_, 0.5 g K_2_HPO_4_∙3H_2_O, 0.1 g KCl, and 0.01 g Ca(NO_3_)_2_ per liter of distilled water. The pH of the medium was adjusted to 3 with 10% sulfuric acid. The medium was autoclaved for 1 h at 120 °C and subsequently cooled at room temperature before use. A filter-sterilized trace-element solution (10 mL) and sulfur powder (1 g), an energy source for SOB, were added to 100 mL medium. The trace-element solution included 11 mg FeCl_3_∙6H_2_O, 0.5 mg CuSO_4_∙5H_2_O, 2.0 mg H_3_BO_4_, 2.0 mg MnSO_4_∙H_2_O, 0.8 mg NaMoO_4_∙2H_2_O, 0.6 mg CoCl_2_∙6H_2_O, and 0.9 mg ZnSO_4_∙7H_2_O in 10 mL distilled water. Oxygen was continuously sparged to the medium to provide an electron acceptor for SOB. Cultivation was performed in a shaking incubator (JSSI-070, JSR, Gongju, Republic of Korea) at 37 °C with 50 rpm mixing intensity. SOB were cultivated for 3–4 d. To provide SOB with comparable activity in subsequent toxicity tests, we evaluated the activity of SOB by measuring changes in electrical conductivity for 1 h before being employed. Only SOB showing an increase in electrical conductivity of 0.10–0.12 mS/cm were used for toxicity tests.

### 2.2. SOB Toxicity Test and Optimization of Test Conditions

The SOB toxicity test kit consisted of a 25 mL flat-bottom glass vial with a rubber stopper and a plastic cap (Figure 1).

SOB toxicity tests were conducted as follows. First, a certain amount of SOB and 5 mL of contaminant-spiked medium were added to the kit. Oxygen was sparged to the headspace of the kit and medium for around 1 min. Then, initial electrical conductivity (in the solution) was measured using an electrical conductivity meter (InLab 737, Mettler Toledo, Columbus, OH, USA). The kit was closed with a cap and rubber stopper and incubated with the vial lying on its side in a shaking incubator for 30 min. After incubation, electrical conductivity was determined again. All SOB toxicity tests were performed in triplicate.

Optimization for test conditions (initial cell density, incubating temperature, and mixing intensity during incubation) of the SOB toxicity kit was performed. We tested 10^5^, 2 × 10^5^, 5 × 10^5^, and 10^6^ cells/mL initial cell densities; 27, 32, 37, and 42 °C incubating temperatures; and 70, 100, 120, and 150 rpm mixing intensities. Hence, a total of 64 combinations of test conditions were evaluated with mercury (0.01, 0.05, 0.1, 0.2, and 0.5 mg/L Hg^2+^). We assessed the detection sensitivity and reproducibility of SOB tests with (30 min) half-effective concentration (EC_50_) and coefficient of variation (CV) for EC_50_ from triplicate kit tests, respectively.

Using the conditions obtained from the above optimization tests, SOB toxicity tests were conducted using heavy metals (Ag^2+^, As^3+^, CN^−^, Cr^6+^, Cu^2+^, Hg^2+^, and Zn^2+^) and petrochemicals (benzene (B), toluene (T), ethylbenzene (E), and *p*-xylene (X), collectively referred to as BTEX) to evaluate improvement in the current technique compared to earlier SOB tests [31,33,34,35].

### 2.3. Chemicals and Laboratory Analyses

All chemicals and sulfur powder used in the present study were ACS grade and had at least 99.9% purity. They were all purchased from Sigma-Aldrich (St. Louis, MO, USA) and employed without further purification. Contaminants (heavy metals and petrochemicals (BTEX)) tested in the current study were prepared according to our earlier studies [31,32,33,35]. Concentrations of heavy metals were calculated based on ions. BTEX were prepared as follows. Each crude liquids of B, T, E, and X was diluted in a nutrient mineral buffer [21] with 0.1% dimethyl sulfoxide twice to create target concentrations of BTEX used in toxicity tests. Amounts of BTEX in the liquid phase of the test kits were determined before and after toxicity testing using high-performance liquid chromatography (HPLC, Water Corporation, Milford, MA, USA). The detailed methodology for using HPLC was presented by Eom et al. (2023) [35].

The toxicity of each contaminant was evaluated by SOB inhibition (%) using Equation (2). As described in this equation, toxic responses of SOB (SOB inhibition) to contaminants were determined by comparisons of increases in electrical conductivity between the controls (the test kits where no contaminant was spiked) and test samples (where specific concentrations of contaminant were spiked).
(2)$$\mathrm{Inhibition}(\%)=(1-\frac{Increase\;in\;electrical\;conductivity\;in\;sample\;\;for\;30\;min\;incubation\;}{Increase\;in\;electric\;conductivity\;in\;control\;for\;30\;min\;incubation})\times 100$$

EC_50_ values for contaminants were determined by the Hillslope equation (Equation (3)). In all dose–response relations, the lowest and highest effects were set to 0% and 100%, respectively.
(3)$$Y=Bottom+\left(\frac{Top-Bottom}{(1+10{)}^{\left(\left(logEC_{50}-X\right)\times Hillslope\right)}}\right)$$
where *X* is the dose of contaminant, *Y* is the toxic response of SOB (SOB inhibition), *Top* is the maximum toxic response, and *Bottom* is the minimum response.

For filter sterilization of the trace-element solution, Nalgene bottle-top sterile filter (0.2 μm) was employed. SOB cell density was determined using a hemocytometer (Paul Marienfeld GmbH & Co. KG, Lauda-Königshofen, Germany). We first measured cell density in the liquid medium and then diluted it to target cell densities. To evaluate statistical significance among data (SOB inhibition and EC_50_ values), ANOVA analysis was performed. A *p*-value of less than 0.05 was seen as statistical significant.

## 3. Results and Discussion

### 3.1. Optimization for Test Conditions of SOB Toxicity Kit

The results (30 min EC_50_ for Hg^2+^ and CV values) from the optimization tests are summarized in Table 1.

Overall, the employment of smaller initial cell densities (10^5^ and 2 × 10^5^ cells/mL) was lower (30 min) EC_50_ values (for Hg^2+^) than the application of larger initial cell densities (5 × 10^5^ and 10^6^ cells/mL) under identical incubating temperatures and mixing intensities. This result indicates that smaller initial cell concentrations yielded more favorable detection sensitivity than larger initial cell concentrations. In microbial toxicity tests, initial cell concentration is a vital factor in determining detection sensitivity because toxicant availability per cell depends on initial cell density [3,41]. Hence, it is expected that employment of smaller initial cell density can lead to improved toxicity detection sensitivity. Lin et al. (2005), Singh and Shrivastave (2015), and Eom et al. (2021) confirmed this advantage (better detection sensitivity) with the employment of smaller initial cell density [3,41,42]. However, it was reported that less initial cell concentration can negatively impact test reproducibility (Lin et al., 2005; Eom et al., 2021) [3,42]. Our data also support this disadvantage (poor test reproducibility) with the application of smaller initial cell density. Particularly, employment of 10^5^ cells/mL of initial cell density resulted in fairly greater CV values than application of 2 × 10^5^, 5 × 10^5^, and 10^6^ cells/mL of initial cell densities. For example, CV values from the tests in which 10^5^, 2 × 10^5^, 5 × 10^5^, and 10^6^ cells/mL of initial cell densities were employed ranged 8.3–12.5, 2.1–4.7, 2.3–5.0, and 1.2–5.0%, respectively. These data show that 10^5^ cells/mL of initial cell density led to poor test reproducibility compared to the other initial cell concentrations. Among the four tested initial cell densities, we chose 2 × 10^5^ cells/mL as the optimal initial cell density achieving favorable performance in both sensitivity and reproducibility. As discussed above, employment of 10^5^ cells/mL of initial cell density resulted in poor test reproducibility, while application of 5 × 10^5^ and 10^6^ cells/mL of initial cell densities caused relatively inferior detection sensitivity (higher EC_50_ values) than 2 × 10^5^ cells/mL of initial cell density.

In terms of incubating temperature, lower temperatures (27 and 32 °C) resulted in more decreased EC_50_ values than higher temperatures (37 and 42 °C). For example, when incubating temperatures were 27, 32, 37, and 42 °C under 2 × 10^5^ cells/mL initial cell density and 120 rpm mixing intensity, (30 min) EC_50_ values (for Hg^2+^) were 44.7, 38.0, 60.3, and 78.3 μg/L, respectively, demonstrating that lower temperatures yielded better detection sensitivity than higher temperatures. We speculate that incubating temperature is associated with microbial activity of SOB. SOB are mesophilic bacteria. There are numerous studies reporting that SOB show active microbial activity up to 40–42 °C [43,44,45]. Our tests also found that as incubating temperatures rose from 27 °C to 42 °C, EC increased in the control tests (where no contaminant was spiked) also escalated, indicating that SOB were more active as the incubating temperatures increased. However, high activity of SOB did not necessarily lead to favorable sensitivity in toxicity detection. SOB with high microbial activity can be less inhibited by the toxicity of contaminants. Therefore, relatively low incubating temperatures (27 and 32 °C), where SOB showed less activity, achieved better sensitivity than higher incubating temperatures (37 and 42 °C). Test reproducibility, on the other hand, was not significantly affected by incubating temperatures. CV values from the tests incubated under 27, 32, 37, and 42 °C were not very different if identical initial cell density and mixing intensity were applied. Among the four tested temperatures, we chose 32 °C as the optimal incubating temperature. EC_50_ values from 27 °C and 32 °C incubating temperatures did not show statistical significance (*p*-value = 0.17). The CV values from 32 °C incubating temperature were slightly lower than those from 27 °C incubating temperature when initial cell densities and mixing intensities were same.

Concerning mixing intensity during incubation, 120 and 150 rpm showed better detection sensitivity than 70 and 100 rpm. For example, (30 min) EC_50_ values (for Hg^2+^) from the tests with 70, 100, 120, and 150 rpm under 2 × 10^5^ cells/mL initial cell density and 32 °C incubating temperature were analyzed at 96.3, 88.3, 38.0, and 44.3 μg/L, respectively. We conjecture that mixing intensity provides SOB increased opportunities for contact with oxygen, which is an e-acceptor of SOB, in the headspace of test kits and with nutrients in the medium. In our tested range, as mixing intensities raised (from 70 rpm to 150 rpm), EC increases in the control tests also escalated. This finding suggests that increased mixing intensity boosts the activity of SOB, which can have a negative impact on detection sensitivity as discussed above. However, mixing intensity can also give SOB the opportunity to interact with contaminants in water, which is a factor contributing to favorable sensitivity. Accordingly, increasing mixing intensity creates both positive and negative impacts on detection sensitivity. Considering EC_50_ data, the positive impact seems to be stronger than the negative impact in our optimization tests. Increased mixing intensity to 120 rpm led to better detection sensitivity. Test reproducibility was not substantially influenced by mixing intensity; four tested different mixing intensities resulted in largely similar CV values if identical initial cell densities and incubating temperatures were employed. Because 120 rpm yielded the lowest EC_50_ values and achieved comparable CV values compared to the others, we considered 120 rpm the optimal mixing intensity.

In summary, from the optimization tests, we determined that 2 × 10^5^ cells/mL initial cell density, 32 °C incubating temperature, and 120 rpm mixing intensity were the optimal test conditions, allowing favorable detection sensitivity and reproducibility in our SOB kit tests.

### 3.2. Comparisons of SOB Toxicity Test Results between the Current Optimal and Earlier Techniques

Using the above optimal test conditions, we conducted toxicity tests for heavy metals (Ag^2+^, As^3+^, CN^−^, Cr^6+^, Cu^2+^, Hg^2+^, and Zn^2+^) and petrochemicals (benzene, toluene, ethylbenzene, and *p*-xylene), and compared the results with data from our previous SOB tests (Table 2).

As shown in Table 2, the current tests resulted in generally lower EC_50_ values compared to earlier ones. For example, our earlier tests [31,32,33,34] obtained 1.76–3.62, 0.20, 4.90, 1.17–2.70, 5.00, 0.21–0.92, and 1.55 mg/L of (2 h) EC_50_ values for Ag^2+^, As^3+^, CN^−^, Cr^6+^, Cu^2+^, Hg^2+^, and Zn^2+^, respectively; however, the current (30 min) EC_50_ values were significantly lower than the earlier data (currently, 0.195, 0.042, 0.673, 0.456, 0.859, 0.038, and 0.692 mg/L were obtained for Ag^2+^, As^3+^, CN^−^, Cr^6+^, Cu^2+^, Hg^2+^, and Zn^2+^, respectively). Furthermore, we previously had 166.1, 94.4, 38.9, and 34.3 mg/L of (24 h) EC_50_ values for benzene, toluene, ethylbenzene, and *p*-xylene, respectively [35]; currently, 35.7, 20.5, 4.0, and 3.7 mg/L of (30 min) EC_50_ values were obtained for benzene, toluene, ethylbenzene, and *p*-xylene, respectively. (After 30 min incubation, BTEX concentrations in the liquid phase of test kits were remained above 94% of initial values, which is in the range of the OECD guidance. Because our test kits were tightly closed and completely sealed with parafilm, it was expected that no BTEX escaped from the test kits. Hence, volatilization of BTEX seems to make no significant impact on EC_50_ values.) These comparisons of EC_50_ values indicate that the current optimized test technique improved sensitivities for toxicity detection compared to our earlier technique. In addition, it is worth noting is that we obtained this improved sensitivity result even with a shorter incubating time (30 min).

The current SOB tests also resulted in favorable test reproducibility. In our earlier SOB tests [34,35], CV values ranged from 8.6% to 12.7%. However, the CV values obtained from the present study were fairly lower (2.4–4.6%) than these earlier values. In addition, Van Coillie et al. (1982) and Blaise et al. (1986) reported that conventional algal toxicity tests accounted for 20–32% of test variability in terms of reproducibility [46,47]. Considering these data, one may conclude that the current SOB technique shows favorable test reproducibility.

### 3.3. Advantages of SOB Toxicity Tests

The present study developed an improved SOB-based toxicity test kit for simple and rapid on-site application. To achieve this, we applied a form of suspended SOB, rather than SOB attached on sulfur particles, decreased the processing time to 30 min, and optimized test conditions in terms of initial cell density, incubating temperature, and mixing intensity. In our earlier tests [2,21,29,30,31,32,33,34,35], SOB attached to the surface of insoluble sulfur particles were used as the test organism. This methodology made it difficult to apply identical amounts of SOB in the toxicity tests. We estimated the amount of SOB by the amount of sulfur particles. Furthermore, in our earlier tests [2,21,29,30,31,32,33,34,35], contact between SOB and contaminants was relatively indirect because SOB resided as a form of biofilm (attached on the surface of insoluble sulfur particles). However, the suspended form of SOB in this study yielded identical amounts of SOB and SOB made direct contact with contaminants.

As previously discussed, initial cell density determines toxicant availability per cell; incubating temperature and mixing intensity are relevant to microbial activity of SOB. By optimizing these conditions, the current SOB tests showed better detection sensitivity and test reproducibility than our previous technique. Furthermore, our SOB tests demonstrate more favorable detection sensitivity even when compared to other existing microbial ecotoxicity tests. For example, Dalzell et al. (2002) reported that toxicity tests using nitrifying bacteria, *Vibrio fischeri*, *E. coli*, and activated sludges resulted in 22.5–37.5 mg/L, 3.7–41.5 mg/L, 0.87–67.5 mg/L of EC_50_ values for Cr^6+^, Cu^2+^, and Zn^2+^, respectively [48]. Cho et al. (2004) showed that *Vibrio fischeri*-based tests resulted in 0.8–1.6 mg/L, 17.2–18.9 mg/L, 12.6–25.2 mg/L of EC_50_ values for Hg^2+^, Cr^6+^, As^3+^, respectively [15], which are much higher than our data. Kudlak et al. (2011) also reported that *Daphnia magna*-based toxicity tests obtained significantly greater EC_50_ values for Zn^2+^ (11.56 mg/L) and Hg^2+^ (9.6 mg/L) than our SOB tests [49].

Our SOB toxicity tests have numerous advantages compared to other microbial ecotoxicity tests. SOB are chemolithoautotrophic bacteria, indicating that SOB do not use organic matters as carbon and electron sources. Moreover, according to our earlier studies [9,34], SOB showed relatively constant activity even under severe conditions. These properties of SOB suggest that SOB are less affected by organic substances and characteristics of test samples. Hence, it is expected that SOB-based tests can be employed in diverse natural environments. In addition, the species of SOB used in the current study is acidophilic bacteria. Generally, contaminated water (by heavy metals or various organic contaminants) has low pH. However, most microbial ecotoxicity tests operate under neutral pH [23,38]. To use these tests, one must adjust pH to neutral. Because pH is a significant factor determining toxicity, such necessary adjustments of pH can inadvertently alter the initial toxicity of the test samples [38,50]. On the other hand, the current SOB tests do not require this pH adjustment; hence, the initial toxicity of test samples is not distorted. Employment of electric conductivity as an end-point measurement is also a merit of our SOB tests. In many microbial toxicity tests (such as bioluminescence bacteria or *Escherichia coli*-based tests), light absorbance is usually used as the end-point measurement [38,51,52]. This parameter is highly affected by the turbidity and color of the test samples, potentially causing false readings of the end-point. However, electric conductivity is independent of these characteristics of the test samples. Consequently, our SOB-based toxicity technique can lead to more accurate test results.

In the present study, we also focused on on-site application. In general, microbial ecotoxicity tests are laboratory-based tests. To measure microbial activity, laboratory analytical instruments are required. Despite the relatively shorter processing time of microbial toxicity tests compared to high trophic level organism-based ecotoxicity tests, several hours of processing of microbial toxicity tests are unsuitable for on-site applications. In contrast, our SOB tests require a simple portable electric-conductivity meter. The necessary processing time of the current SOB tests is only 30 min. Moreover, we developed portable kit-type toxicity tests for better mobility. These advantages make the current SOB toxicity test technique suitable for on-site applications. However, in spite of these numerous advantages, our SOB tests still need a portable incubating system for field applications. We are now developing this system.

Our future research includes further verification of the reliability of the current SOB toxicity tests with more diverse contaminants. In addition, we will employ this SOB test to evaluate biochemical and mixture toxicity of contaminants in various environments, including the hydrosphere, geosphere, and atmosphere.

## 4. Conclusions

The current study aimed to improve an existing SOB-based toxicity test technique for simple and rapid on-site application. To achieve this goal, we developed a 25 mL glass-vial-based toxicity kit test. We employed a form of suspended SOB and decreased the processing time to 30 min. To yield favorable detection sensitivity and test reproducibility, we determined the optimal test conditions of the SOB toxicity kit to be 2 × 10^5^ cells/mL initial cell density, 32 °C incubating temperature, and 120 rpm mixing intensity. Compared to the test results from our earlier SOB technique, the current technique resulted in lower EC_50_ values from toxicity tests of diverse contaminants, such as heavy metals, inorganic nitrogen, endocrine-disrupting compounds, and petrochemicals. It also showed favorable CV values. Our SOB kit test has a number of advantages, such as no need for advanced analytic instruments and no distortion of test results from characteristics of test samples, making it suitable for simple and rapid on-site application.

## Figures and Tables

**Figure 1 toxics-11-00352-f001:**
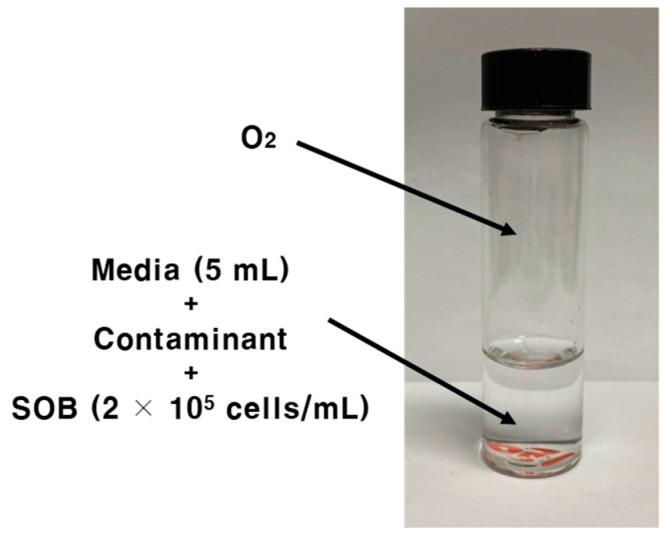
A SOB toxicity test kit. This 25 mL glass-vial-based SOB kit consists of 2 × 10^5^ cells/mL of initial cell density and 5 mL of media. Headspace and media are purged with oxygen. Incubation is conducted in a shaking incubator with 120 rpm mixing intensity at 32 °C for 30 min.

**Table 1 toxics-11-00352-t001:** Results from optimization for test conditions of SOB toxicity kit (30 min EC_50_ for Hg^2+^ and CV values, depending on test conditions).

Initial Cell Density (Cells/mL)	Incubating Temperature (℃)	Mixing Intensity (rpm)	EC_50_ (μg/L)	CV (%)	Initial Cell Density (Cells/mL)	Incubating Temperature (°C)	Mixing Intensity (rpm)	EC_50_ (μg/L)	CV (%)
10^5^	27	70	90.7	12.1	5 × 10^5^	27	70	115.7	4.3
		100	84.7	12.5			100	99.3	5.0
		120	44.7	12.3			120	56.4	4.5
		150	48.3	11.4			150	64.7	4.7
	32	70	92.0	9.7		32	70	124.0	2.4
		100	85.3	8.3			100	114.1	2.6
		120	44.0	9.1			120	58.0	3.0
		150	44.7	8.5			150	62.3	4.0
	37	70	143.0	10.3		37	70	155.0	3.2
		100	115.3	10.3			100	133.3	3.0
		120	57.3	11.3			120	72.3	3.5
		150	60.0	10.0			150	69.7	4.4
	42	70	154.3	10.1		42	70	174.7	2.3
		100	124.7	10.2			100	147.0	2.4
		120	75.1	11.9			120	86.3	5.7
		150	72.3	10.2			150	88.0	5.0
2 × 10^5^	27	70	93.3	4.3	10^6^	27	70	142.3	3.3
		100	84.0	3.1			100	126.7	2.5
		120	44.7	4.6			120	67.7	2.3
		150	46.1	3.8			150	74.7	5.1
	32	70	96.3	2.2		32	70	146.0	3.1
		100	88.3	2.8			100	130.3	1.9
		120	38.0	2.6			120	70.7	2.9
		150	44.3	2.6			150	72.3	2.1
	37	70	144.7	3.8		37	70	160.0	1.7
		100	120.3	2.1			100	145.7	4.0
		120	60.3	2.5			120	82.0	3.7
		150	62.1	3.2			150	79.7	1.9
	42	70	165.3	2.4		42	70	187.0	2.8
		100	133.7	4.3			100	159.1	4.7
		120	78.3	2.7			120	92.7	1.2
		150	76.0	4.7			150	94.0	4.6

**Table 2 toxics-11-00352-t002:** Comparisons of results from SOB toxicity tests between the current and earlier studies.

		Earlier SOB Tests				Current SOB Tests		
Contaminant		Processing Time (h)	EC_50_ (mg/L)	CV (%)	Reference	Processing Time (h)	EC_50_ (mg/L)	CV (%)
Heavy metal	Ag^2+^	2	1.76–3.62	-	Gurung et al. (2015) [31]; Ahmed et al. (2018) [33]	0.5	0.195	3.1
	As^3+^	2	0.2	11.5	Eom et al. (2019) [34]	0.5	0.047	4.5
	CN^−^	2	4.9	12.7	Eom et al. (2019) [34]	0.5	0.676	3.3
	Cr^6+^	2	1.17–2.7	10.5	Qambrani et al. (2016) [32]; Ahmed et al. (2018) [33]; Eom et al. (2019) [34]	0.5	0.456	3.0
	Cu^2+^	2	5	-	Ahmed et al. (2018) [33]	0.5	0.860	2.4
	Hg^2+^	2	0.21–0.92	8.7	Ahmed et al. (2018) [33]; Eom et al. (2019) [34]	0.5	0.038	2.6
	Zn^2+^	2	1.55	-	Ahmed et al. (2018) [33]	0.5	0.692	3.4
Petrochemical	Benzene	24	166.1	9.8	Eom et al. (2023) [35]	0.5	35.849	4.6
	Toluene	24	94.4	9.5	Eom et al. (2023) [35]	0.5	20.575	3.8
	Ethylbenzene	24	38.9	9.7	Eom et al. (2023) [35]	0.5	4.038	4.1
	p-Xylenes	24	34.3	8.6	Eom et al. (2023) [35]	0.5	3.803	2.4

## Data Availability

The data presented in this study are available upon request from the corresponding author.

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
