# Peer review of "Development of an Improved Sulfur-Oxidizing Bacteria-Based Ecotoxicity Test for Simple and Rapid On-Site Application"

_toxics, 2023, doi:10.3390/toxics11040352_

Round 1

Reviewer 1 Report

The overall work is an interesting study. However, I have some questions that should be addressed.

I think that the test is very much dependent on the viability of the cells. This in turn is very much influenced by the environmental conditions in the field, such as temperature. How stable is the test over time? How reproducible are the results in the field with different batches of cultures? Does the test require special storage for field applications?

Author Response

Responses to Reviewer 1’s Comments

The overall work is an interesting study. However, I have some questions that should be addressed.

Response: I appreciate Reviewer 1’s positive remarks and valuable comments on our study. Our detailed responses to Reviewer 1’s comments are presented below.

I think that the test is very much dependent on the viability of the cells. This in turn is very much influenced by the environmental conditions in the field, such as temperature. How stable is the test over time?

Response: In the current toxicity tests, we used SOB with comparable activity. Before being empoloyed in toxicity tests, we evaluated activity of SOB by measuring changes in electrical conductivity for 1 h. Only SOB showing an increase in electrical condutivity of 0.10-0.12 mS/cm were used for toxicity tests. In addition, our SOB kit tests were conducted in a shaking incubator wehre constant mixing intensity and temperature were maintained. SOB showed constant activity during test processing. Hence, test results were not differently influenced by test conditions. We have clarified this discussion at lines 115-118 in the revised manuscript.

How reproducible are the results in the field with different batches of cultures?

Response: Because SOB having comparable activity were employed in toxicity tests, we obtained favorable results in terms of test reproducibility. As presented in the original manuscript, CV values from our SOB tests (under optimal test conditions) ranged in 2.4-4.6 %, demonstrating that they were very reproducible.

Does the test require special storage for field applications?

Response: Our SOB tests have numerous advanateges for on-site application. For example, our SOB tests adopt a kit-type reactor that is easy to use in field. Moreover, our SOB tests require only 30 min of processing time and do not need advanced analytic instrument. However, our SOB tests still need some portable incubating system for field application (Currently, we are developing this system). We have supplemented this discussion at lines 337-339 in the revised manuscript.

Reviewer 2 Report

Review of the manuscript

Development of an improved sulfur-oxidizing bacteria-based ecotoxicity test for simple and rapid on-site application “

Author: Heonseop Eom

Dear author of the above manuscript: below I try to answer the questions to increase the readability of the review

General remarks

What is the main point of the paper?

Increasing speed and ease to test aquatic samples with sulfur-oxidizing bacteria

Title

Does the title represent the aims and conclusions? Yes

Is the title accurate? Yes

Content

Are the aims clear, and does the research address the aims? Yes and No

Does the writing stick to that point? Not always

Is the manuscript accurate? Yes

Is the manuscript concise? Not always

Abstract

Two terms are not really explained in the abstract: first in line 10: ...„improved…“

and Line 19 …“...better test results…“. It would help I think if the author would give a statement about what kind of improvement was accomplished.

Introduction

line 31ff: the authors writes as if only the exceedence of a level of concern is seen as water contamination. I would like to argue that any chemical/substance which is introduced by humans is a contamination per se. It might not be a known risk to humans or other known test organisms but that does not mean that 1. it is not a contamination and 2. that it still might affect organisms (yet unknown until today). So I would ask the author to restructure that sentence if he agrees.

Line 50ff: The authors states the arguments for the usage of bacteria. I would rather add an information that beside all the points stated, bacteria in addition are by far more or at least as important than/as other organisms in terms of ecological functions (destruents etc. etc.).

Line 74ff: change to: ...“to enable on-site field application“…? Or … on-site and field application? As I am not a native speaker, too the author might ask somebody else

The last parts of the introduction read more like a discussion. as also some part of the discussion seem to fit better to the introduction

What I think is missing is a comparison to the standard test in ecotoxicology with other e.g. luminescent bacteria and their sensitivity for example to metals. This seems important to me as the other bacteria are more often used in risk assessment (I think) and the introduction of a new bacteria tests needs to prove any improvement to exiting tests. As speed and volume of tests and easyness of testing is an improvement the comparison to other bacterial tests in terms of sensitivity is missing in the discussion

Methods

The methods are well written. But I would put the sentence of purity of cultivation chemicals further up in the text (under paragraph 2.1 and not 2.3). Also add the company where the sulfur powder was bought including the purity. Please also add any company and type of machinery which was used (e.g. the shaking incubator etc.).

Please state if the shaking of the vials was done with vial standing or lying on the side.

What kind of filter was used to sterilize the trace element solution? How was the growth of the culture checked (standard OD at xyz nm?).

The heavy metals which were used: please indicate the salts used (I assume that the concentrations given are calculated in terms of the Hg2+? And not the salt ?

Results

The tables are not completely visible. In the pdf they are cut on the right side.

The parameters of all dose response curves are missing. Are the highest effects always set to 100% and the lowest to 0% in the Hill -model? If so please indicate. If not, please add all parameter (min, max, slope and EC50) so that other can use the data further (model mixtures for example).

Line 154: I would not call it: „… a p-value less than 0.5 represents statistical significance…“. As using the p-value of 0.5 or any other is always under discussion I would rather write: „… a p-value less than 0.5 here is seen as statistical significant…“

Line 237 ff: the author states that the sensitivity was „fairly lower“. It seems that the EC50 are about a factor of 10 lower. This I would not just call „fairly“ but that is a significant increase in sensitivity!

Line 240 ff: the benzene etc. are highly volatile, so please state if the test vials are completely gas tight as the substance may decrease in concentrations within minutes. With that in mind it is feasable to assume that the EC50 is much higher with longer exposure i.e. standing i.e. volatilization time. So the author should compare his data with other bacterial assays.

Discussion

see my comments in the introduction part.

Literature

not visible in my pdf file

Global impression:

A good paper suitable for the journal, but with some small shortcomings (especially the comparison to already existing data of other bacterial assays in the literature

This report is interesting and is worthy of consideration. However, I feel that the paper would increase in quality if the author manage to rewrite the introduction and discussion

Recommendation:

Accept after minor revisions

Author Response

Responses to Reviewer 2’s Comments

General remarks

What is the main point of the paper?

Increasing speed and ease to test aquatic samples with sulfur-oxidizing bacteria

Title

Does the title represent the aims and conclusions? Yes

Is the title accurate? Yes

Content

Are the aims clear, and does the research address the aims? Yes and No

Does the writing stick to that point? Not always

Is the manuscript accurate? Yes

Is the manuscript concise? Not always

Response: I appreciate the valuable comments provided by Reviewer 2. Our detailed responses to Reviewer 2’s comments are presented below.

Abstract

Two terms are not really explained in the abstract: first in line 10: ...„improved…“

and Line 19 …“...better test results…“. It would help I think if the author would give a statement about what kind of improvement was accomplished.

Response: The terms ‘improved’ and ‘better test results’ mean that the current SOB tests achieved more favorable toxicity detection sensitivity (lower EC50) and test reproducibility (lower CV) compared to earlier SOB tests. I have revised lines 14 and 22 in the abstract.

Introduction

line 31ff: the authors writes as if only the exceedence of a level of concern is seen as water contamination. I would like to argue that any chemical/substance which is introduced by humans is a contamination per se. It might not be a known risk to humans or other known test organisms but that does not mean that 1. it is not a contamination and 2. that it still might affect organisms (yet unknown until today). So I would ask the author to restructure that sentence if he agrees.

Response: I appreciate this review comment. I partially agree and disagree with this opinion. As Reviewer 2 mentioned, chemicals can have unknown adverse effects on environment and ecology. However, chemicals sometimes serve as essential nutrients for organisms. For example, some heavy metals such as iron, cobalt, manganese, and zine are necessary for metabolism of organisms. When the concentrations exceeds certain levels, even these essential heavy metals act toxic. I have slightly revised lines 32-33 for better clarity.

Line 50ff: The authors states the arguments for the usage of bacteria. I would rather add an information that beside all the points stated, bacteria in addition are by far more or at least as important than/as other organisms in terms of ecological functions (destruents etc. etc.).

Response: A change is made as suggested. I have supplemented importance of microorganisms in terms of ecological functions in the revised manuscript (lines 57-58).

Line 50ff: Line 74ff: change to: ...“to enable on-site field application“…? Or … on-site and field application? As I am not a native speaker, too the author might ask somebody else

Response: I have removed ‘field’ from this sentence. The terms of ‘on-site’ and ‘field’ are redundant words.

The last parts of the introduction read more like a discussion. as also some part of the discussion seem to fit better to the introduction

What I think is missing is a comparison to the standard test in ecotoxicology with other e.g. luminescent bacteria and their sensitivity for example to metals. This seems important to me as the other bacteria are more often used in risk assessment (I think) and the introduction of a new bacteria tests needs to prove any improvement to exiting tests. As speed and volume of tests and easyness of testing is an improvement the comparison to other bacterial tests in terms of sensitivity is missing in the discussion

Response: I appreciate this review comment for better clarity of our manuscript. In the revised manuscript, I tried to provide better introduction and discussion. Some parts in the introduction have moved to the discussion. In particular, lines 80-88 in the original manuscript seem to better fit to discussion. In addition, as suggested, I have supplemented comparisons of detection sensitivity between the current SOB tests and other existing microbial toxicity tests in the revised discussion (lines 299-308).

Methods

The methods are well written. But I would put the sentence of purity of cultivation chemicals further up in the text (under paragraph 2.1 and not 2.3). Also add the company where the sulfur powder was bought including the purity. Please also add any company and type of machinery which was used (e.g. the shaking incubator etc.).

Response: All chemicals and sulfur powder used in this study were ACS grade and have at least 99.9 % of purity. They were all purchased from Sigma-Aldrich and used without further purification. I have clarified this information at lines 149-151 in the revised manuscript. In addition, I have provided information of machinery (shaking incubator and HPLC) used in this study (lines 114 and 158).

Please state if the shaking of the vials was done with vial standing or lying on the side.

What kind of filter was used to sterilize the trace element solution? How was the growth of the culture checked (standard OD at xyz nm?).

Response: SOB test kits were incubated with vial lying on the side. For filter sterilization of the trace element solution, Nalgene bottle-top sterlie filter (0.2 um) was employed. SOB cell density was determined using a hemocytometer. Cell density in the liquid medium was measured first and then diluted it to target cell densities. These explainations have been supplemented at lines 133 and 170-173 in the revised manuscript.

The heavy metals which were used: please indicate the salts used (I assume that the concentrations given are calculated in terms of the Hg2+? And not the salt ?

Response: Concentrations of heavy metals were calcuated based on ions. Chemicals were prepared according to our earlier studies (Eom et al., 2019; Eom et al., 2023).

Results

The tables are not completely visible. In the pdf they are cut on the right side.

The parameters of all dose response curves are missing. Are the highest effects always set to 100% and the lowest to 0% in the Hill -model? If so please indicate. If not, please add all parameter (min, max, slope and EC50) so that other can use the data further (model mixtures for example).

Response: In the revised manuscript, I have resized the tables to fit the page. In all response-dose relations, the lowest and highest effects were set to 0 % and 100 %, respectively (line 165-167).

Line 154: I would not call it: „… a p-value less than 0.5 represents statistical significance…“. As using the p-value of 0.5 or any other is always under discussion I would rather write: „… a p-value less than 0.5 here is seen as statistical significant

Response: A change was made as Reviewer 2 suggested in the revised manuscript.

Line 237 ff: the author states that the sensitivity was „fairly lower“. It seems that the EC50 are about a factor of 10 lower. This I would not just call „fairly“ but that is a significant increase in sensitivity!

Response: I also agree with such improvement is significant. I have changed ‘fairly’ to ‘significantly’ in this sentence.

Line 240 ff: the benzene etc. are highly volatile, so please state if the test vials are completely gas tight as the substance may decrease in concentrations within minutes. With that in mind it is feasable to assume that the EC50 is much higher with longer exposure i.e. standing i.e. volatilization time. So the author should compare his data with other bacterial assays.

Response: Basically, our SOB test kits were gas-tighted. BTEX were prepared according to our earlier study (Eom et al., 2023). Brifely, each of crude liquids of B, T, E, and X was diluted in nutrient mineral buffer (Oh et al., 2011) with 0.1 % of dimethyl sulfoxide twice to create target concentrations of BTEX used in toxicity tests. Concentrations of BTEX in the liquid phase of test kits were measured before and after toxicity tests using HPLC (Eom et al., 2023). After 30 min incubation, BTEX concentrations were remained above 94 % of initial values, which is in the range of the OECD guidance. Since our test kits were tightly closed and completely sealed with parafilm, it is expected that no BTEX escaped from the test kits. Hence, volatilization of BTEX seems to make no significant impact on test results, particularly on EC50 values. We have clarified this discussion at lines 268-272 in the revised manuscript.

Discussion

see my comments in the introduction part.

Response: We have addressed this comment above.

Literature

not visible in my pdf file

Response: The revised manuscript includes references.

Global impression:

A good paper suitable for the journal, but with some small shortcomings (especially the comparison to already existing data of other bacterial assays in the literature

This report is interesting and is worthy of consideration. However, I feel that the paper would increase in quality if the author manage to rewrite the introduction and discussion

Recommendation:

Accept after minor revisions

Response: I appreciate Reviewer 2’s positive review on this study. I have tried to modified our manuscript according to Reviewer 2’s comments.